# An Ultra-Sensitive and Multifunctional Electronic Skin with Synergetic Network of Graphene and CNT

**DOI:** 10.3390/nano13010179

**Published:** 2022-12-30

**Authors:** Yu Wang, Tian-Rui Cui, Guang-Yang Gou, Xiao-Shi Li, Yan-Cong Qiao, Ding Li, Jian-Dong Xu, Yi-Zhe Guo, He Tian, Yi Yang, Tian-Ling Ren

**Affiliations:** 1School of Integrated Circuit, Tsinghua University, Beijing 100084, China; 2Beijing National Research Center for Information Science and Technology (BNRist), Tsinghua University, Beijing 100084, China; 3Center for Flexible Electronics Technology, Tsinghua University, Beijing 100084, China

**Keywords:** electronic skin, graphene, SWCNT, physiological signals detection, artificial throat

## Abstract

Electronic skin (e-skin) has attracted tremendous interest due to its diverse potential applications, including in physiological signal detection, health monitoring, and artificial throats. However, the major drawbacks of traditional e-skin are the weak adhesion of substrates, incompatibility between sensitivity and stretchability, and its single function. These shortcomings limit the application of e-skin and increase the complexity of its multifunctional integration. Herein, the synergistic network of crosslinked SWCNTs within and between multilayered graphene layers was directly drip coated onto the PU thin film with self-adhesion to fabricate versatile e-skin. The excellent mechanical properties of prepared e-skin arise from the sufficient conductive paths guaranteed by SWCNTs in small and large deformation under various strains. The prepared e-skin exhibits a low detection limit, as small as 0.5% strain, and compatibility between sensitivity and stretchability with a gauge factor (GF) of 964 at a strain of 0–30%, and 2743 at a strain of 30–60%. In physiological signals detection application, the e-skin demonstrates the detection of subtle motions, such as artery pulse and blinking, as well as large body motions, such as knee joint bending, elbow movement, and neck movement. In artificial throat application, the e-skin integrates sound recognition and sound emitting and shows clear and distinct responses between different throat muscle movements and different words for sound signal acquisition and recognition, in conjunction with superior sound emission performance with a sound spectrum response of 71 dB (f = 12.5 kHz). Overall, the presented comprehensive study of novel materials, structures, properties, and mechanisms offers promising potential in physiological signals detection and artificial throat applications.

## 1. Introduction

Flexible electronic skin (e-skin) has attracted significant attention due to its diverse potential applications, including in physiological signals detection [1,2,3], health monitoring [4,5], artificial throat [6], and so on. As the vital component of connectivity between the human and intelligent equipment, the respective movement and physiological activities are quantified into electrical signals by attaching the e-skin to the body. To better meet the increasing requirements of multiple application scenarios, the requirements for the e-skin are more stringent, involving accurate and wide-range signal acquisition, conformality, low cost, long-term stability, and multifunction [7,8]. Up to now, much ongoing effort has been made to enhance the performance of the e-skin to sense mechanical signals. The ultrahigh stretchability of 1000 [9] and superior sensitivity of Gauge factor (GF) >2000 [10] have been demonstrated. High-quality signal detection requires incompatibility between sensitivity and stretchability as well as sufficient adhesion to the body [11]. Many e-skins are prepared on a substrate and encapsulated using elastomeric polymers. Due to poor adhesion, additional tape is often required to attach the e-skin to the skin [12,13], and the non-tight interface reduces the transfer of mechanical deformation to the sensing layer, decreasing the quality of signals. On the other hand, as a significant component of the physiological signal, the acoustic signal has a prospective application in the artificial throat. Many e-skins based on mechanical stimuli sensing have shown great potential in sound vibration detection due to the low detection limit and ultra-sensitivity [14,15,16]. Meanwhile, research on the sound emitting performance of these flexible e-skins and the integration of reorganization of sound recognition and emission is significant in the artificial throat application. Therefore, an ideal multifunctional integrated e-skin, with incompatibility between sensitivity and stretchability, and close attachment to the skin to detect high-quality signals, is urgently needed.

As an advanced carbon material, graphene has been used as the sensing layer in the e-skin due to its unique two-dimensional (2D) layered structure with superior electronic properties and mechanical flexibility [17,18]. The satisfied e-skin with incompatibility between sensitivity and stretchability needs to ensure large structural deformation resulting in a large amount of resistance change, while also having sufficient conductive paths under large deformations under strain [19]. Therefore, embedding a cross-linked conductive network in a 2D nanosheet is an efficient strategy to optimize the sensitivity and stretchability of e-skin [11,20,21,22]. As a one-dimensional (1D) material, CNT is a promising candidate due to its high electrical conductivity, large aspect ratio, and favorable strengths [23,24,25]. Furthermore, a substrate attached tightly to the human body is of paramount importance. PU with self-adhesive properties is feasible for the substrate due to its high stretchability, biocompatibility, and high adhesion. The performance of self-adhesion ensures close contact between the e-skin and body, so the deformation can be completely transferred to the sensing layer.

In this work, a sandwich structure e-skin based on the SWCNTs crosslinked multilayer graphene (SCG) was successfully fabricated by directly preparing the sensing layer on the flexible PU and encapsulating it with PU thin-film tape. The synergistic crosslinked SWCNTs within and between the graphene layers were produced during the sonication between graphene ink and SWCNTs aqueous dispersion, which ensures sufficient conductive paths. The prepared e-skin provides a low detection limit as small as 0.5% strain, a large sensing range as high as 60% strain, and a high sensitivity with the GF of 964 at a strain of 0–30%, and 2743 at a strain of 30–60%. In the duration test, the performance of the e-skin was not significantly degraded after 1000 repeated stretching/releasing experiments. For physiology signals detection, the e-skin can detect subtle motions, such as artery pulse, as well as large body motions, such as knee joint bending, elbow movement, and neck movement. Furthermore, when used as the artificial throat, there are clear and distinct responses between different throat muscle movements and different words for sound signal acquisition and recognition, in conjunction with superior sound emitting performance with a sound spectrum response of 71 dB (f = 12.5 kHz) due to the ultrasmall heat capacity of the PU thin-film tape substrate. This work shows that ultrahigh sensitive, superior stretchable, and multifunctional SCG e-skin offers promising potential in physiological signal detection and artificial throat applications.

## 2. Materials and Methods

### 2.1. Materials

A 5 wt% graphene ink with a sheet diameter of 1–5 μm and SWCNTs ink with a length of 5–30 μm, a diameter of 1–2 nm, and a purity greater than 90% were provided by XFNANO Materials Tech Co. Ltd. (Nanjing, China). The ethylene glycol was purchased from Titan Scientific Co. Ltd. (Shanghai, China). The medical PU tape was purchased from HONS Medical Tech Co. Ltd. (Shanghai, China), which is a Class 1 Medical Device, and the in vitro cytotoxicity test, skin sensitization test, and skin irritation test all passed professional tests. All the chemical reagents were used without further treatment.

### 2.2. Fabrication of the SCG E-Skin

Graphene ink, SWCNT ink, and ethylene glycol were mixed at a volume ratio of 5:1:1 and then sonicated for 20 min under the power of 200 W to produce a uniform suspension. After that, the mixed solution was dropped onto the PU thin film and patterned with polyimide masks (40 × 5 mm^2^ rectangular pattern size), then placed into the fume cupboard for about 12 h at room temperature to form the SCG/PU thin film. After dripping silver paste on both ends of the SCG/PU thin film, conductive copper tape is pasted to form two electrodes. Then, the sensor was encapsulated by another PU thin-film tape. Due to the hard texture of silver paste, there is no strain in the electrode contact part of the device under strain application.

### 2.3. Characterization

The surface and the cross-section morphology of the SCG were employed by scanning electron microscope (SEM, Zeiss merlin, Oberkochen, Federal Republic of Germany). The Raman spectroscopy (LabRAM HR Evolution, HORIBA Jobin Yvon, Japan) was performed using a laser with a wavelength of 532 nm and a power of 50 mW at room temperature. The electromechanical property of the SCG e-skin was performed with a universal testing machine (Shimadzu AGS-X, Shimadzu, Kyoto, Japan) coupled with a digital source meter (Rigol DM 3068, RIGOL TECHNOLOGIES CO.,LTD, Suzhou, China). The test system diagram was shown in Appendix A. During the test, the electrode part of the e-skin is clamped by the fixture, and the strain is applied to change the resistance of the middle part of the e-skin. The SPL of the SCG e-skin was measured by the standard microphone (Superlux ECM999, Goang-Fann Co., Ltd., New Taipei City, Taiwan), and the swept-frequency signal from 100 Hz to 20 kHz was generated by the dynamic signal analyzer (Agilent 35670A, Agilent Technologies, Inc., California, CA, USA).

## 3. Results and Discussion

### 3.1. Design and Fabrication of SCG E-Skin

Figure 1 summarizes the fabrication and characteristics of the SCG e-skin. Figure 1a schematically illustrates the procedure of the sandwich-like SCG e-skin, which is low in manufacturing cost, simple, and biocompatible. Firstly, the uniform synergistic network of graphene sheets and SWCNTs was prepared by sonicating. Then, the mixed solution was dropped onto a PU thin-film tape. In this study, PU thin-film tape contains a release paper layer, a PU layer with adhesion on one side, and a substrate layer. Using the adhesive PU layer of e-skin involves peeling off the release paper layer, dripping the SCG solution on the PU layer, and attaching the e-skin to the body surface after releasing the substrate layer of PU thin-film tape when used. The strong adhesion of the e-skin generates a tight connection between the e-skin and skin surface, and the deformation is completely transferred to the sensing layer, producing high-fidelity and ultra-sensitive sensing. After preparing two electrodes with the conductive copper tape and silver paste, the e-skin was encapsulated by another PU thin-film tape as the support PU tape layer. For the intuitive understanding of the SCG e-skin, the digital picture of the SCG e-skin is shown in Appendix A. For comparison and optimization, the SCG e-skin with different graphene and SWCNTs ink ratios, such as 1:1, 10:1, 20:1, and 100% graphene, were fabricated by the same method. Figure 1b shows the cross-section illustration of the sandwich-structure SCG e-skin, consisting of the SCG sensing layer, support PU tape layer, and adhesive PU tape layer. The strain is applied on two sides of the e-skin. The Raman spectra is a common demonstration method to indicate the quality of materials. Figure 1c shows the Raman spectra of the SWCNTs, graphene, and SCG film. For graphene thin film, the D-peak around 1350 cm^−1^ and the G-peak around 1585 cm^−1^ are the typical characteristic peaks of graphene [26]. Furthermore, the G-peak and the 2D-peak (around 2700 cm^−1^) indicate that the graphene sheet film has several layers. Additionally, the radial breathing modes (RBMs) around 186 cm^−1^, D-peak around 1347 cm^−1^, G^+^-peak around 1567 cm^−1^, and G^−^-peak around 1593 cm^−1^ demonstrate the SWCNTs film [27]. By comparison, the as-synthesized graphene/SWCNT thin films display the RMB around 186 cm^−1^, which proves the presence of SWCNTs. Furthermore, the almost similar D/G band ratios to graphene thin film prove the presence of graphene sheets. Combined, it indicates the absolute assembly of graphene and SWCNTs with no damage, and the effectiveness of the fabrication. Figure 1d–g presents the scanning electron microscope (SEM) image of the surface morphology and SCG film cross-section. The top-view SEM image (Figure 1d,e) shows that the film is randomly stacked and overlapped with the graphene sheets. When strain is applied to the SCG, cracks are caused due to the sliding between the graphene sheets, and the long hairy SWCNTs are bridging between the cracked graphene sheets. The cross-section SEM images (Figure 1f-g) demonstrate that the SCG film is a multilayered structure, and the long hairy SWCNTs within graphene layers were observed. In this way, the synergistic crosslinked SWCNTs between the graphene sheets and graphene layers keep adequate electron transmission paths upon a large strain, thus resulting in a large sensing range and ultra-high sensitivity. Therefore, the high adhesive PU substrate and the sandwich structure e-skin based on the SWCNTs crosslinked multilayer graphene simultaneously endow the SCG e-skin with high-quality signal recording, large stretchability, and high sensitivity at the same time.

### 3.2. Mechanical Performance of SCG E-Skin

The desired e-skin needs to meet the requirement of superior performance to ensure large resistance variation under the strain tension, which can be measured by the GF, which is defined as:GF = (ΔR/R_0_)/(ΔL/L_0_),(1)
where ΔR and R_0_ represent the variation in the resistance under the different strains and the initial resistance under no strain, and ΔL and L_0_ represent the variation of length and the initial length. Figure 2a illustrates the two linear parts, 0–30%, and 30–60%, with extracted GF values of 964 and 2741. At 0–30% strain, there is a slow increase in resistance variation, indicating the formation of cracks caused by the relative sliding between the multilayer graphene layer. As the applied strain increased to 30–60%, the rapidly increasing resistance variation was due to the expansion of the cracks and reduced conductive pathways, compared with the e-skin based on pure graphene, which shows the GF of 260 at a strain of 0–30%, and 856 at a strain of 30–60%. The high sensitivity is due to the synergistic crosslinked SWCNTs, which ensure an adequate electron transmission path in small and large deformation under various strains. In Figure 2b, the resistance variation under tiny strain (0.5–5%) was measured with the frequency of 0.25 Hz, which is 1.25 under 0.5% strain, 2.60 under 1% strain, 11.0 under 3% strain, and 19.0 under 5% strain. Such high sensitivity under tiny strain ensures the detection of subtle motions, such as artery pulse and sound vibration. Different frequency responses under 10% strain are illustrated in Figure 2c, it is obvious that the SCG e-skin exhibits a stable frequency response at the relatively wide frequency range of 0.1–1.0 Hz, and the peak value of the resistance variation at different frequencies is almost the same, which may be due to the good adhesion between the substrate and the SCG sensing layer. Figure 2d depicts no significant degradation in performance after 1000 cyclic stretching/releasing experiments under the 10% strain with the frequency of 1.0 Hz. Thus, the long-term stability and durability make the SCG e-skin suitable to be used in practical applications. For comparison and optimization, the SCG e-skin with a different mixture ratio of graphene and SWCNTs (1:1, 5:1, 10:1, and 15:1), and the pure graphene e-skin (mixing ratio of Graphene and CNT is 0 in Figure 2e) were prepared following the same method, and the responses to the strain of 0.5–5% were presented in Figure 2e. The resistance variation at each strain increased first and then decreased with the increase in graphene ratio, and the highest response occurs when the volume ratio is 5:1. In conclusion, the SCG e-skin exhibits the excellent mechanical performance of large stretchability, high sensitivity in the working range, low detection limit, and stability, which is superior to previous findings (Figure 2f) [28,29,30,31,32,33,34,35,36].

### 3.3. Relative Resistance Change Mechanism

The SCG e-skin can detect deformation by sensing resistance variation, and a microstructure model of the SCG under different strains is proposed to explain the working mechanism (Figure 3a). In the first stage, the sensing film has a randomly stacked and overlapped structure, and the sliding between adjacent graphene sheets dominates the resistance variation. As the applied strain increases, the area of the overlapping decreases, causing the contact resistance to increase. In the second stage, as the applied strain increases, the overlapped graphene sheets separate. Therefore, the number and the size of cracks increases. In this stage, the long hairy SWCNTs bridged between the cracked graphene sheets guarantee an adequate electronic transmission path. The resistance variation is dominated by the synergistic crosslinked SWCNTs between the graphene sheets and the tunneling resistance between graphene components. Figure 3b shows the resistance model of the SCG e-skin under different applied strains, which could provide deeper understanding.

### 3.4. SCG E-Skin for Physiological Signals Detection

The fancy materials and structure give the SCG e-skin conformal lamination, a large workable strain range, and high sensitivity, which facilitate its application in physiological signals detection. Among the various physiological signals of the human body, the pulse plays a vital role in reflecting the body’s condition [37,38]. Due to the self-adhesive of the PU thin film, the SCG e-skin is attached to the radial artery without additional assistance, and the pulse signals can be captured accurately, including three differential peaks, the percussion wave (P-wave), tidal wave (T-wave), and the diastolic wave (D-wave) (Figure 4a,b). Under normal conditions, the heart rate is calculated to be 72 beats min^−1^, which is within the normal heart rate range of healthy people. Moreover, the SCG e-skin is attached next to the eye to monitor blinking (Figure 4c), indicating that the SCG e-skin can detect muscle movements caused by tiny facial expressions, which has a potential application in 3D human−machine interactions in the future [39]. In addition to the tiny signal, the SCG e-skin is used to detect large human activities. As shown in Figure 4d, the SCG e-skin can detect the movement of neck flexion and make a good distinction between slight bending and heavy bending. This is crucial for people to sit properly to protect the cervical spine. Furthermore, the small and large bending movements of the elbow joint are also investigated, showing distinguishing, stable, and repeatable responses (Figure 4e). Figure 3f shows that the resistance variation rapidly increases as the knee joint bending increases, and returns to the initial value when the knee joint returns to its original position. Such diverse physiological signals reflect our condition, and the SCG e-skin will become a new paradigm for health diagnosis, health care, and fitness tracking [40,41].

### 3.5. SCG E-Skin for Sound Recognition and Emitting

The artificial throat, as an essential wearable device to help dumb people “speak”, needs to combine the reorganization of sound vibration, detection of throat muscle movement, and sound emission [42]. Due to the excellent mechanical properties of the SCG e-skin, a high-quality and high-fidelity detection of sound vibrations as well as throat muscle movements is ensured. When fixed above the commercial loudspeaker, the SCG e-skin has a good sound detection ability. As shown in Figure 5a and Appendix A, the animal’s sound waves can be reconstructed well. Besides that, individual words can also be recognized due to their obvious characteristic waves (Figure 5b and Appendix A). When attached to the throat, the responses of different throat muscle movements, including coughing (Figure 5c), making an “ah” tone (Figure 5d), nodding, and shaking head (Appendix A) can be recognized. From the results, it can be concluded that different throat muscle movements can be identified and recorded due to the distinguished characteristics peak. Moreover, it can identify the different sound vibrations of the word “lemon”, “pineapple”, “peach”, and “apple” (Figure 5e). All results demonstrate that the ultrahigh sensitivity and large workable range of SCG e-skin have good prospects for sound vibration recognition and throat muscle movement detection.

For the sound emitting needs of the artificial throat, the low thermal conductivity of the PU thin-film substrate decreasing heat leakage, combined with the high thermal and electronic conductivity of the SCG, will improve the efficiency of thermoacoustic conversion based on thermoacoustic conversion acoustic devices. Sound pressure level (SPL) is the most commonly used indicator of acoustic wave strength and correlates well with human perception of loudness [43]. Another important indicator is sound pressure (SP), which represents the change in pressure caused by sound waves. Figure 6a illustrates the schematic diagram of the test system. When AC voltage is applied to the device, Joule heating is generated by alternating currents and causing periodic vibrations on the surface air of the device, and then sound responses are collected and analyzed by the standard microphone and dynamic frequency analyzer. The microphone is placed 0.5 cm above the device to research the relationship between the SPL and different input power. As seen in Figure 6b, the SPL gradually increased as the input power increased from 0.23 W to 1.2 W. The increased input power causes the device to generate more Joule heat, and the surface air molecules are heated up to form the vibration, resulting in a higher sound pressure level. The highest SPL is 72 dB (f = 12.5 Hz) when the input power is 1.2 W. Figure 6c shows the responses of output SP with input power at different frequencies (f = 5 Hz, 7.5 Hz, 10 Hz, 12.5 Hz). With increasing input power, the output sound pressure increases linearly when the frequency is certain. The responses at different test distances are also displayed in Figure 6d. Longer distances results in lower and uneven sound pressure levels. The relationship of the SP and the test distance at different frequencies is shown in Figure 6e, and the sound pressure decreases in inverse proportion with the distance increases. Finally, Figure 6f demonstrates that the device has no degradation after 120 min of continuous testing (tested every 5 min). Long-term stability and durability mean that the SCG e-skin can be used in practical applications.

The adhesion of the PU thin-film substrate generates a tight attachment between the throat and the SCG e-skin, and the high sensitivity of the sensing layer enables the electronic skin to detect different sounds and throat muscle movements accurately. Meanwhile, the excellent efficiency of thermoacoustic conversion guarantees the high performance of sound emission. Therefore, the integration of authentic detection and distinguished sound emission of SCG e-skin has the potential application in the artificial throat.

Herein, we fabricated a versatile e-skin, based on the synergistic network of graphene and SWCNTs, which exhibits excellent properties with a low detection limit as small as 0.5% strain, compatibility between sensitivity and stretchability, and the application of physiological signals detection and artificial throat was also demonstrated. Despite the better results achieved, there are still many areas to explore and challenge, which will also be studied in our future work. First of all, for practical application, due to the complexity and variability of the actual environment, the feasibility and useful lifetime in practical application scenarios should be tested and improved, especially in the presence of various external noises. Secondly, the influence of environmental humidity and human sweat to the e-skin is very important. Therefore, it is essential to study the response of e-skin to humidity for the expansion of multiple application scenarios. Thirdly, the regulation of carbon nanotubes and the influence of strain on the microstructure are still lacking at the microscopic level. Combined with the research at the theoretical level, it will help us to improve the performance of the device better and explain the microscopic mechanism.

## 4. Conclusions

In this work, we develop an SWCNTs-crosslinked multilayer graphene e-skin, which is ultrahigh sensitive, has superior stretchability, and is multifunctional. The PU tape substrate with self-adhesion offers a tight attachment to the skin, which guarantees that the deformation is completely transferred to the sensing layer. The multilayer structure of graphene makes it easy to slide, and the synergistic-crosslinked SWCNTs within and between graphene layers guarantee the adequate electron transmission path in large deformations under strain, and such a novel structure achieves high sensitivity in each workable range. In physiology signals detection, the SCG e-skin could be used to realize the full range of body motion detection, ranging from subtle deformation (blinking, pulse signal) to large joint movement (neck, elbow, knee bending). Furthermore, when used as an artificial throat, the SCG e-skin integrates sound recognition and sound emission. This superior performance, multifunctional, integrated electronic skin is of great significance in the field of intelligent diagnosis, health care, and artificial throat applications in the future.

## Figures and Tables

**Figure 1 nanomaterials-13-00179-f001:**
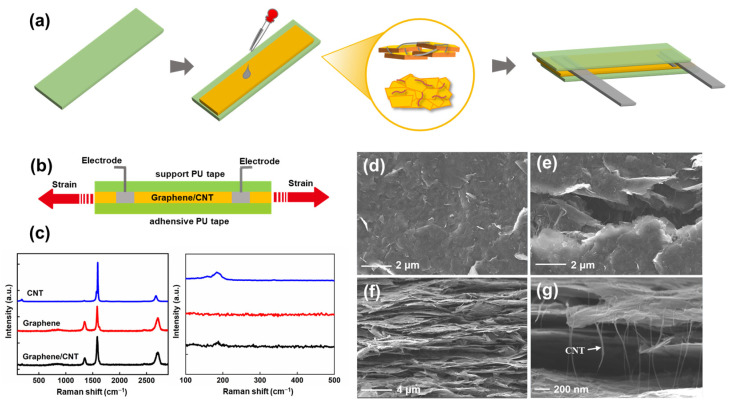
Fabrication and characteristics of the SCG e-skin. (**a**) The fabrication process of SCG e-skin. (**b**) Cross-section illustration of a single SCG e-skin. (**c**) Raman spectra of SWCNT, Graphene, and Graphene/SWCNT, in which the right picture shows the characteristic peak at 100–500 cm^−1^. (**d**) Top-view SEM of the SCG film. (**e**) Top-view SEM of the crack in the SCG film. (**f**) Cross-sectional SEM images of SCG film. (**g**) Cross-sectional SEM images of SCG film with high magnification.

**Figure 2 nanomaterials-13-00179-f002:**
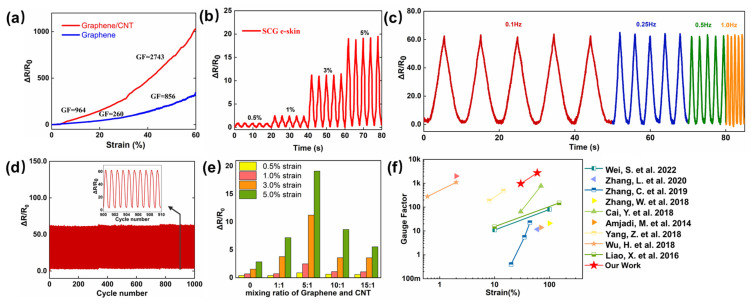
Electromechanical performance of SCG e-skin. (**a**) Relative resistance variation of SCG e-skin and graphene e-skin. (**b**) Relative resistance variation under different cyclic strains at a frequency of 0.25 Hz. (**c**) Relative resistance variation of SCG e-skin under cycle stretching–releasing with a strain of 10% at frequencies of 0.1, 0.25, 0.5, and 1.0 Hz. (**d**) Relative resistance variation with the cyclic strain of 10% for 1000 cycles. (**e**) The maximum resistance variation of different mixture ratios of graphene and SWCNTs (1:1, 5:1, 10:1, 15:1, and pure graphene) under the strain of 0.5–5%. (**f**) Comparison of sensitivity-strain characteristics with others [28,29,30,31,32,33,34,35,36].

**Figure 3 nanomaterials-13-00179-f003:**
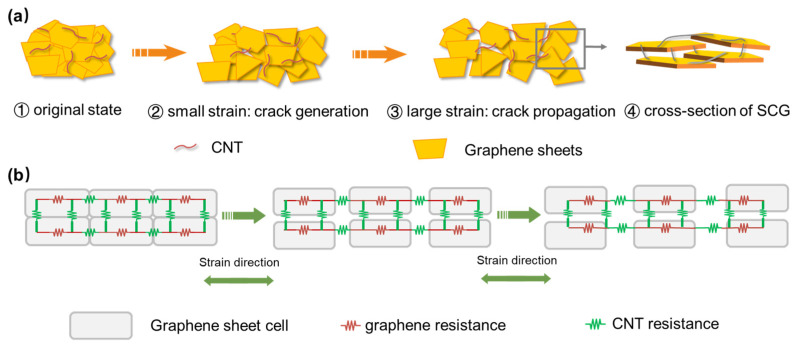
The relative resistance variation mechanism of the Graphene/CNT e-skin. (**a**) Layered stacked graphene flacks with CNTs crosslinked e-skin on original state, and generation and propagation of the crack under small and large strains. The arrow indicates the schematic diagram of the cross-section. (**b**) The resistance model of the SCG e-skin under different strains.

**Figure 4 nanomaterials-13-00179-f004:**
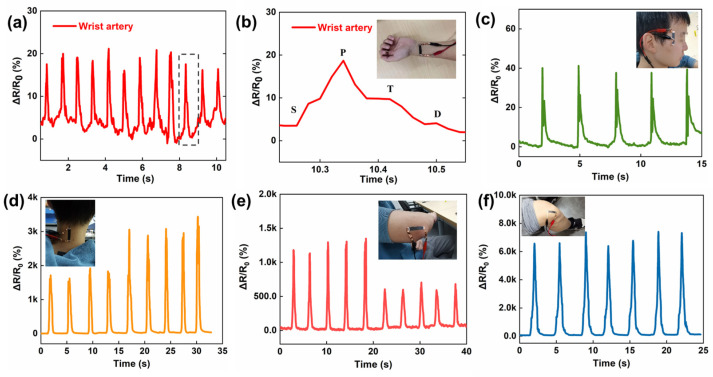
Mechanical physiological signals detected by SCG e-skin. (**a**) Pulse signals detected by the SCG e-skin on the wrist artery. (**b**) The zoomed-in graph of a single pulse signal. Inset: tester with the SCG e-skin attached to the wrist. (**c**) Blinking detected by the SCG e-skin attached next to the eye. Inset: tester with the SCG e-skin attached next to the eye. (**d**) Small and large range of neck movement detected by the SCG e-skin. Inset: tester with the SCG e-skin attached to the neck. (**e**) Small and large range of bending movement of elbow detected by the SCG e-skin. Inset: tester with the SCG e-skin attached to the elbow. (**f**) Bending movement of the knee joint detected by SCG e-skin. Inset: tester with the SCG e-skin attached to the knee.

**Figure 5 nanomaterials-13-00179-f005:**
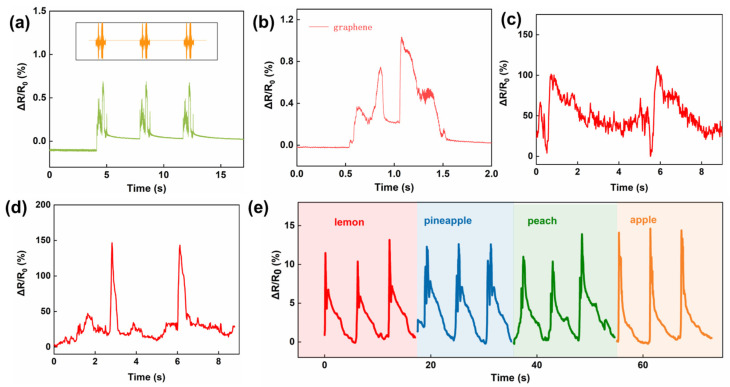
Sound and throat movement detected by SCG e-skin. (**a**) The comparison between the initial audio wave of a chirp and the corresponding sound vibration signal. (**b**) The response to the English word “graphene” from a loudspeaker. (**c**) Throat muscle movement of cough detected by SCG e-skin attached to the throat. (**d**) Throat muscle movement makes an “ah” tone detected by SCG e-skin attached to the throat. (**e**) Responses toward the sound “lemon”, “pineapple”, “peach”, and “apple” when the device is attached to the throat.

**Figure 6 nanomaterials-13-00179-f006:**
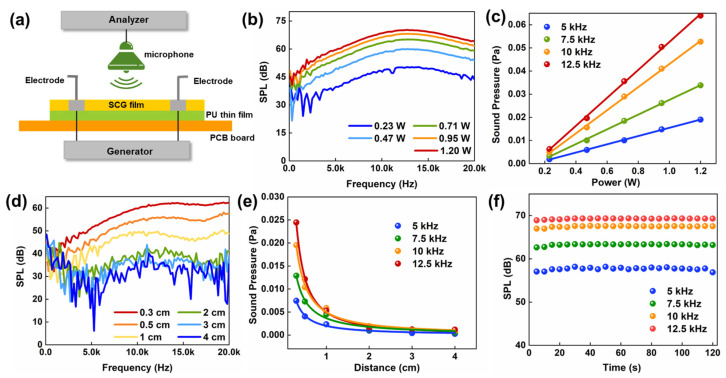
The performance of emitting sound. (**a**) Schematic structure of the device and the schematic diagram of the test platform. (**b**) The SPL responds to different powers. (d = 0.3 cm). (**c**) The output SP versus the input power at 5, 7.5, 10, and 12.5 kHz. (**d**) The SPL response at different test distances. (**e**) The output SP versus the test distance at 5, 7.5, 10, and 12.5 kHz. (**f**) The stability of the output SPL for 120 min and each test for 5 min measurements at 5 kHz, 7.5 kHz, 10 kHz, and 12.5 kHz.

## Data Availability

The data presented in this study are available within the article.

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
