# Peer review of "An Ultra-Sensitive and Multifunctional Electronic Skin with Synergetic Network of Graphene and CNT"

_nanomaterials, 2022, doi:10.3390/nano13010179_

Round 1
Reviewer 1 Report
In the work titled “An Ultra-Sensitive and Multifunctional Electronic Skin with Synergetic Network of Graphene and CNT” authors presented solid results on the fabrication and testing of sensitive devices based on PU, SWCNTs and graphene. They prepared an e-skin for physiological signals detection applications. The work brings interesting results in the context of e-skin, pointing out the synergetic effect of SWCNTs with graphene. However, the novelty of the work is not very high (see Chem. Nano. Mat. 2021,7, 982–99 or Adv. Funct. Mater.2017,27, 160606, for instance).
I recommend this manuscript for publication in Nanomaterials after addressing major and minor issues.
1. Authors show beautiful drawings of the developed device, but they must provide some digital pictures of the real e-skin, also when it is being applied for monitoring physiological signals.
2. Authors provide very interesting results about the electromechanical performance of the e-skin in Figure 2, and for the physiological signals detection in Figure 4. However, Time is always seconds in the exe x. A considerable comment must be done, at least, regarding the stability and the useful life-time of this e-skin (apart from Relative resistance variation with the cyclic strain of 10% 173 for 1000 cycles in Figure 2).
3. In lines 187-189, the authors state that " The high sensitivity is due to the synergistic crosslinked SWCNTs, which ensure adequate electron transmission path in small and large deformation under various strains". However, some controls, such as graphene-only or SWCNT-only based electronic skin, are missing to really see the synergistic effect.
4. Most of the references in the bibliography are from Asiatic authors. It would be highly desirable to include more works by authors from different origins.
5. Titles of sections 3.2 (line 168) and 3.3. (line 208) are similar.
6. What do the authors mean with “convenient” in line 132? Have the authors tested and demonstrated the biocompatibility of e-skin, apart from a few seconds during physiological signal tests?
7. Raman characterization (lines 148-152) should be better explained and improved. “Raman spectrums” should be replaced by “Raman spectra” (line 147).
8. “Strain sirection” must be replaced by “Strain direction” in Figure 3.
Reviewer 2 Report
Submitted manuscript touched on a modern and relevant topic e-skin fabrication and characterization. Studies and measurements were carried out at the proper level, and results presented in clear and logical form. High number of the repetition cycles and wide range of applied strain (0 to 60%) makes prerequisites for use in practical applications.
The paper was written in logical and clear for and the size of the manuscript is adequate. However, there are couple of mistypes
Line 128. Top-view SEM of the creak in the SCG film. Probably crack. Mistype
Figure 4 sirection instead of direction. Mistype.
The methodology was chosen quite correctly, however, to my opinion some point may be improved. First of all, it is better to use 4-contact set up for resistance measurements. From presented results is not clear how strain application can modify SCG-SWCNT/ copper interface and thereby change the contact resistance. Four-point measurement should not be sensitive to contact resistance change.
Also, despite of statement about “non-destructive performance after 1000 cyclic experiments” (line 197) it’s useful to compare SEM images of cross sections before (it is presented in manuscript) and after repeated strain application, it there are any structural changes.
One more thing, analysis of influence of humidity to e-skin properties is also crucial for possibility of practical plications. So it may by useful to publish it in future .
Nevertheless, to my opinion, presented results are very interesting and manuscript can be published in present form after minor corrections. In general, the presented experimental results are interesting which may be used practical applications.
Round 2
Reviewer 1 Report
Almost all changes have been taken into account by the authors, but the English spelling of the main text and figures still needs to be checked (i.e. "spectrums", "sirection", etc...).
